# Increased In Vivo Exposure of N-(4-Hydroxyphenyl) Retinamide (4-HPR) to Achieve Plasma Concentrations Effective against Dengue Virus

**DOI:** 10.3390/pharmaceutics15071974

**Published:** 2023-07-18

**Authors:** Alexander J. Martin, David M. Shackleford, Susan A. Charman, Kylie M. Wagstaff, Christopher J. H. Porter, David A. Jans

**Affiliations:** 1Nuclear Signalling Laboratory, Department Biochem. & Mol. Biol., Biomedicine Discovery Institute, Monash University, Clayton 3800, Victoria, Australia; alexander.martin@monash.edu (A.J.M.); kylie.wagstaff@monash.edu (K.M.W.); 2Centre for Drug Candidate Optimisation, Monash Institute of Pharmaceutical Sciences, Monash University, 381 Royal Parade, Parkville 3052, Victoria, Australia; david.shackleford@monash.edu (D.M.S.); susan.charman@monash.edu (S.A.C.); 3Drug Delivery Disposition and Dynamics, Monash Institute of Pharmaceutical Sciences, Monash University, 381 Royal Parade, Parkville 3052, Victoria, Australia; chris.porter@monash.edu

**Keywords:** N-(4-hydroxyphenyl) retinamide, fenretinide, dengue virus, pharmacokinetic analysis, cytochrome P450 metabolism, lipid formulation, SEDDS (self-emulsifying drug delivery systems)

## Abstract

N-(4-hydroxyphenyl) retinamide (4-HPR, or fenretinide) has promising in vitro and in vivo antiviral activity against a range of flaviviruses and an established safety record, but there are challenges to its clinical use. This study evaluated the in vivo exposure profile of a 4-HPR dosage regime previously shown to be effective in a mouse model of severe dengue virus (DENV) infection, comparing it to an existing formulation for human clinical use for other indications and developed/characterised self-emulsifying lipid-based formulations of 4-HPR to enhance 4-HPR in vivo exposure. Pharmacokinetic (PK) analysis comprising single-dose oral and IV plasma concentration-time profiles was performed in mice; equilibrium solubility testing of 4-HPR in a range of lipids, surfactants and cosolvents was used to inform formulation approaches, with lead formulation candidates digested in vitro to analyse solubilisation/precipitation prior to in vivo testing. PK analysis suggested that effective plasma concentrations could be achieved with the clinical formulation, while novel lipid-based formulations achieved > 3-fold improvement. Additionally, 4-HPR exposure was found to be limited by both solubility and first-pass intestinal elimination but could be improved through inhibition of cytochrome P450 (CYP) metabolism. Simulated exposure profiles suggest that a b.i.d dosage regime is likely to maintain 4-HPR above the minimum effective plasma concentration for anti-DENV activity using the clinical formulation, with new formulations/CYP inhibition viable options to increase exposure in the future.

## 1. Introduction

RNA viruses constitute an immense threat to human health worldwide, as highlighted by the SARS-CoV-2 pandemic, which has caused more than 200 million infections and 4.2 million deaths as of August 2021 [1]. Other key human pathogens include human immunodeficiency virus-1 (HIV-1), which currently affects 37.7 million people and has killed approximately the same number since the start of the epidemic [2], and influenza A virus, which, apart from pandemics such as the 1918 “Spanish flu” that caused 50 million deaths [3], continues to be a threat every year, despite being kept in check by a yearly vaccine approach in many countries. Finally, mosquito-borne infections by flaviviruses, which include dengue virus (DENV), Zika virus (ZIKV), West Nile virus (WNV), Japanese encephalitis virus (JEV) and yellow fever virus (YFV), account for a growing number of yearly infections; DENV alone causes more than 400 million infections every year [4], whereas a large outbreak of >750,000 infections by ZIKV spanned 70 countries in 2015–2016 [5,6], with this number having grown to 87 countries with evidence of ZIKV transmission to date [7].

Viruses exploit multiple components of the infected host cell in order to replicate efficiently and spread to infect new host cells. Importantly, all of the above viruses exploit members of the host importin (IMP) superfamily to transport a viral component into the nucleus, in most cases to subvert the host immune response [8,9]. DENV/ZIKV/WNV/JEV/YFV viruses, in particular, co-opt the host nuclear import-mediating IMPα/β1 heterodimer to transport viral non-structural protein 5 (NS5); in the case of DENV, viral attenuation can be achieved by mutations preventing NS5 nuclear import [10], whilst small molecule inhibitors such as ivermectin targeting IMPα can inhibit infection by DENV/ZIKV/WNV [11,12,13,14]. Ivermectin can also inhibit infection by SARS-CoV-2 [15,16], influenza A [17], HIV-1 [11], and other RNA viruses (e.g., Venezuelan encephalitis virus [18]).

We identified the synthetic retinoid N-(4-hydroxyphenyl) retinamide (4-HPR) through high-throughput screening (HTS) as a specific inhibitor of the binding of DENV NS5 to IMPα/β1 [19]. Additionally, 4-HPR exhibited EC50 values of c. 1 μM against all four infectious DENV strains in vitro, but also in an ex vivo human peripheral blood mononuclear cell model of severe (antibody-dependent enhanced–ADE) infection [19]. Furthermore, 4-HPR has since been shown to be efficacious against ZIKV [20], WNV (Kunjin) [21,22] and YFV [23]. Most importantly, a lethal mouse model of severe ADE DENV infection showed protection of 70% of infected mice through b.i.d (twice daily) administration with 20 mg/kg 4-HPR [19]; however, no pharmacokinetic (PK) analysis was performed to inform the potential exposure required to achieve this important therapeutic outcome, and the 4-HPR formulation used is not suitable for human use.

Although 4-HPR has an established safety record from testing in Phase I and II clinical trials for various cancer indications, its relatively poor aqueous solubility remains a challenge [24]. The standard human clinical formulation (the “HC formulation”) is a suspension of 4-HPR in corn oil with polysorbate 80 as a surfactant [25], which can achieve plasma levels of up to 12.9 μM at a dose of 4000 mg/m^2^/day in patients [26]; patient compliance is a significant issue due to the large dose volumes required. As such, there is a clear need for an optimised formulation with enhanced solubility and in vivo PK properties.

In this study, we set out to evaluate the in vivo exposure profile of the 4-HPR formulation (mouse dengue efficacy formulation—MDE) shown to be effective in severe DENV infection [19] and compare it to the profile of the HC formulation that is approved for human clinical use for other indications. Our PK analysis indicates that effective plasma concentrations can be achieved in mice with the HC formulation, but simulated exposure profiles indicate that trough concentrations are below the apparent therapeutic threshold in a b.i.d administration regime. We subsequently build on this to enhance 4-HPR in vivo exposure properties, developing and characterising novel self-emulsifying lipid-based formulations of 4-HPR, which show >3-fold improved exposure over the HC formulation in mice. We find that 4-HPR exposure was limited by both solubility and first-pass intestinal elimination but could be improved through inhibition of cytochrome P450 (CYP)-medicated intestinal metabolism. The results overall suggest that although the HC formulation achieves plasma 4-HPR levels above the predicted minimum effective plasma concentration for anti-DENV activity in a mouse model, self-emulsifying lipid-based formulations and CYP inhibition represent viable future options to increase exposure.

## 2. Materials and Methods

### 2.1. Materials

First, 4-HPR (C_26_H_33_NO_2_—see Figure 1—GM Pharma, FL, USA), diazepam (Sigma, St. Louis, MO, USA), 1-aminobenzotriazole (ABT; Honeywell Fluka, International Inc., Charlotte, NC, USA) were purchased from the sources indicated previously [19]. The excipients used in formulation experiments were sourced as indicated; Capryol 90, dimethylsulphoxide (DMSO), Gelucire 44/14, Labrasol, Lauroglycol 90, Maisine 35-1, Maisine CC and Transcutol were from Gattefossé (Saint-Priest Cedex, France); Capmul MCM EP and Captex 355 EP/NF from ABITEC Corporation (Columbus, OH, USA); benzyl alcohol, corn oil, hydroxypropyl methylcellulose, polyethylene glycol (PEG) 400, polysorbate 80, RPMI-1640, soybean oil and Tween 85 from Merck KGaA (Darmstadt, Germany); Cremophor EL, Kolliphor RH40 from BASF SE (Ludwigshafen, Germany); ethanol from Ajax-Finechem, Scoresby, Australia); and foetal bovine serum (FBS) from CellSera Australia (Rutherford, Australia).

For in vitro digestion experiments, lipoid E PC S (phosphatidylcholine) was from Lipoid GmbH (Ludwigshafen, Germany), 4-bromophenylboronic acid, sodium taurodeoxycholate, porcine pancreatin, sodium chloride, sodium hydroxide, and Trizma maleate from Merck KGaA (Darmstadt, Germany), and calcium chloride dihydride from BDH Chemicals Australia (Kilsyth, Australia).

Ultrapure water was obtained from a Milli-Q purification system (Merck KGaA, Darmstadt, Germany). All other solvents were of HPLC grade (Merck KGaA, Darmstadt, Germany).

### 2.2. In Vivo PK Analysis

Studies with non-fasted male C57Bl/6 mice were conducted using established procedures in accordance with the Australian Code of Practice for the Care and Use of Animals for Scientific Purposes; study protocols were reviewed and approved by the Monash Institute of Pharmaceutical Sciences Animal Ethics Committee.

C57Bl/6 mice (22.6–28.7 g) having access to food and water ad libitum throughout the pre- and post-dose sampling period were dosed with 4-HPR either orally by gavage (2–10 mL/kg, depending upon formulation) or IV by bolus injection into the lateral tail vein (50 μL/animal, 2 mL/kg), and blood samples were collected up to 49 h post-dose by either submandibular bleed or terminal cardiac puncture (*n* = 3–5 mice per timepoint for each formulation) with a maximum of four samples from each mouse. Blood samples were mixed with heparin, potassium fluoride and complete protease inhibitor cocktail (Roche Diagnostics, Mannheim, Germany), centrifuged and supernatant plasma was removed and stored at −80 °C until analysis by LC-MS. Specifics of formulations used in PK experiments are detailed in Appendix A.

For in vivo CYP inhibition experiments, repeat oral doses of ABT (50 mg/kg b.i.d commencing 4 h before 4-HPR administration, 5 mL/kg by gavage) were administered according to a regimen shown to maintain ongoing CYP inhibition [27]. Blood samples were collected from ABT-treated mice for up to 72 h after dosing 4-HPR.

For 4-HPR quantification, proteins were precipitated from plasma samples using acetonitrile (2-fold volume ratio) prior to analysis using a Waters Acquity Ultra-High Performance Liquid Chromatography (UHPLC) system coupled to a Waters Xevo TQ MS system (Waters Corporation, Milford, MA, USA). Chromatographic separation was performed using a C8 reverse phase column (Ascentis Express RP C8, 50 × 2.1 mm, 2.7 μm, Merck KGaA, Darmstadt, Germany) and a 3 μL injection volume. Gradient elution (methanol-water gradient with 0.05% formic acid) was performed at a flow rate of 0.4 mL/min, cycle time of 4 min, with detection by positive electrospray ionisation in multiple-reaction monitoring mode. Then, 4-HPR standards (10,000–0.5 ng/mL) were prepared in blank plasma, with diazepam as an internal standard (10 μL of 5 μg/mL in 50% acetonitrile/water per sample). For 4-HPR, the retention time was 2.97 min, *m*/*z* transition 392.23 > 161.15, cone voltage 30 V and CID 20 V.

### 2.3. PK Data Analysis and Profile Simulations

The dose administered to each mouse was calculated on the basis of the pre-dose body weight, the volume of formulation administered, and the 4-HPR concentration in the formulation. The plasma concentration versus time profile was defined by the average plasma concentration at each sample time, and PK parameters were calculated using non-compartmental methods (PKSolver Version 2.0) using the average dose administered to the dosing group.

For statistical analysis of exposure data, the 95% z-confidence interval for the dose-normalised AUC_0–30h_ was calculated in R Statistical Software (v4.1.2; R Core Team 2021) via Bailer’s method [28], implemented as “batch” design in the package “PK” (Version 1.3.5) [29]. Only data up to 30 h post-exposure were used for statistical comparisons to ensure that statistical comparisons were not biased by differences in the timeframe over which exposure data was monitored.

To illustrate the differences in 4-HPR plasma exposure profiles that would be expected in mice upon repeat-dose administration (b.i.d, 10 h/14 h at 20 mg/kg) of the different formulations, profiles were simulated using a simple compartmental PK approach. Initially, a linear 2-compartment model was fitted (PKSolver, Version 2.0) to the mean plasma concentration-time profile obtained after bolus IV administration of 4-HPR. The parameter estimates thus obtained (i.e., V1, k10, k12 and k21—see Appendix A) were considered to define the systemic disposition of 4-HPR and were fixed during simulation of the oral profiles. Oral profiles were simulated using a differential equation-based compartmental model scripted in Berkeley Madonna (version 8.3.1.8). Systemic disposition parameters were fixed as described above, under the assumption that the post-absorption distribution and elimination of 4-HPR was independent of the formulation administered. Bioavailability was fixed at the value determined by non-compartmental analysis, and the first order rate constant for compound absorption was manually varied until the simulated concentration-time profile was in close agreement with the experimental concentration-time data obtained after single-dose administration of pre-dispersed lipid formulation. As all single-dose oral exposure profiles exhibited a similar Tmax, this value for the absorption rate constant was used for the simulation of all repeat-dose exposure profiles. These were obtained by setting the bioavailability (F) to the value determined for each formulation by non-compartmental analysis.

### 2.4. Solubility Studies

Excipients were weighed directly into glass vials at the indicated concentrations, mixed thoroughly, then transferred to Eppendorf tubes and loaded with an excess of 4-HPR. Formulations were then heated to 37 °C, vortexed extensively (initially every hour) and agitated with a stirrer and incubated at 37 °C in a shaking incubator. Periodically (every 1–3 days), undissolved drug was removed from the formulation by centrifugation (20 min at 2100× *g*, 37 °C) and 20 mg of supernatant was transferred to new tubes for quantification by UHPLC (stored at −20 °C), with the remaining formulation vortexed and agitated vigorously incubated at 37 °C in a shaking incubator until the next sampling. Samples were dissolved in 1 mL 1:1 chloroform:methanol, then diluted 1:100 in acetonitrile.

In this case, the UHPLC assay used a Shimadzu Nexera X2 UHPLC system (Shimadzu Corporation, Kyoto, Japan), with chromatographic separation on a C18 reverse phase column (Kinetex 2.6 μm C18 100 Å, LC Column 50 × 2.1 mm, Phenomenex Inc., Torrance, CA, USA) at 30 °C, with samples stored at 10 °C in the autosampler until injection (3 μL volume). Isocratic elution (79.95% acetonitrile, 19.95% H_2_O, 0.1% formic acid) was performed at a flow rate of 0.3 mL/min, with absorbance measured by the photodiode array detector. Then, 4-HPR standards (50–0.521 μg/mL) were created by successive serial dilution of a 50 μg/mL stock standard in acetonitrile, with area under curve at absorbance of 365 nm used to generate a standard curve using LabSolutions software V.5.82.

### 2.5. Plasma Protein Binding

Protein binding of 4-HPR was assessed in human (pooled, Innovative Research Inc., Novi, MI, USA) and mouse (collected in-house from male C57BL6 mice) plasma via Rapid Equilibrium Dialysis (RED) using pre-saturated dialysis units as previously [30]. Briefly, RED inserts were pre-saturated overnight with 400 ng/mL 4-HPR in PBS, after which solutions were removed and discarded, and 300 µL aliquots of diluted human or mouse plasma (10% *v*/*v* in pH 7.4 PBS) containing 4-HPR (2000 ng/mL) were added to the donor chamber of the RED inserts (*n* = 4/matrix) and dialysed against 500 µL of PBS (containing 40 ng/mL 4-HPR). During the 24 h dialysis period, the RED system was sealed with a gas impermeable film, and the plate was maintained at 37 °C under ambient atmosphere in an orbital plate shaker set at 800 rpm (ThermoMixer C; Eppendorf, Hamburg, Germany). At the end of the dialysis period, aliquots were taken from each donor and dialysate chamber to obtain post-dialysis measures of the total and unbound 4-HPR concentrations, respectively. The measured pH of pre- and post-dialysis donor matrix and dialysate buffer was within pH 7.4 ± 0.1. In parallel to the binding assessment, stability of 4-HPR in each diluted plasma matrix was assessed; there was no evidence of any degradation.

Donor and dialysate samples were processed and analysed using a matrix matching approach, and all samples were stored frozen at −80 °C until analysis by LC-MS. Compound binding was assessed on the basis of the measured concentrations in dialysate and donor samples at the end of the dialysis period, assuming that the system was at steady state by 24 h. As the binding assay was performed using diluted plasma, data were corrected for the dilution factor to give a binding value for neat plasma via an established approach which accounts for the shift in equilibria that occurs with protein dilution [31].

## 3. Results

### 3.1. PK Analysis of the MDE Formulation in Mice Indicates an Effective In Vivo trough Concentration of c. 1 μM 4-HPR

As indicated above, Fraser et al. [19] identified 4-HPR by HTS as an agent targeting dengue protein NS5 and demonstrated its potential using the MDE formulation to protect against DENV in a mouse model of severe (ADE) infection using a b.i.d administration protocol. However, no PK analysis was performed to determine the therapeutic concentration achieved in plasma. We initially decided to address the question of the 4-HPR exposure profile in the study by performing PK analysis for single-dose oral administration of the formulation in conjunction with exposure modelling for b.i.d administration. In parallel, we compared this to the exposure profile of the standard human clinical “HC formulation” [25].

PK analysis for systemic in vivo exposure of 4-HPR was performed in non-fasted male C57Bl/6 mice dosed orally with the MDE and HC formulations, as well as a standard comparator aqueous suspension vehicle (HPMC-SV) (see Appendix A). Then, 4-HPR was rapidly absorbed when given orally in HPMC-SV (Figure 1A), but the apparent oral bioavailability was revealed to be only 4% (Table 1). Absorption was also rapid when given as the MDE formulation, but compared to the aqueous suspension, exposure was notably higher, and the apparent bioavailability increased to approximately 13%, suggesting a possible solubilising effect of serum in the formulation. The higher plasma exposure of 4-HPR after dosing both the MDE and HPMC-SV formulations, compared with HC, was deemed to be statistically significant based on the lack of overlap of the 95% z-confidence intervals for the dose-normalised AUC_0–30h_ of each formulation (see Appendix A).

**Figure 1 pharmaceutics-15-01974-f001:**
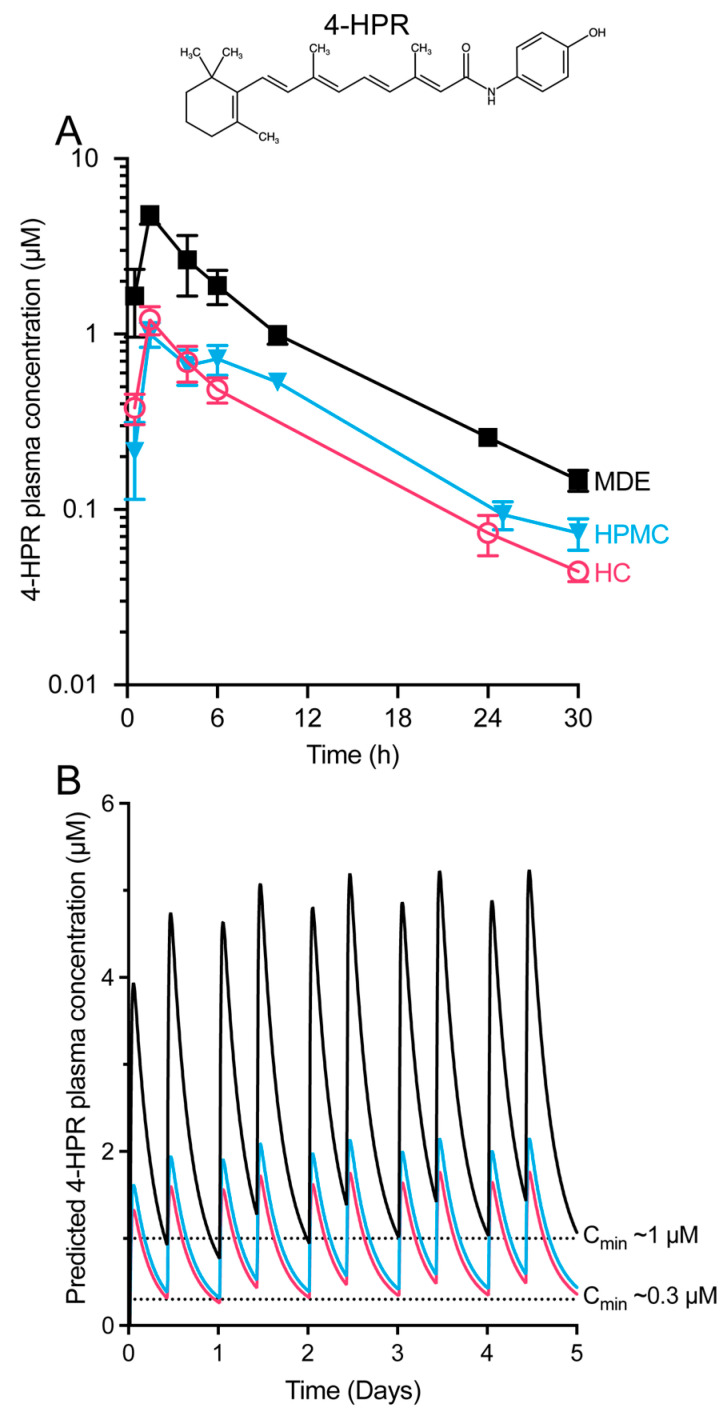
PK analysis (**A**) and simulated b.i.d exposure profiles (**B**) indicate that an effective concentration of 4-HPR (structure shown above) (**A**) is likely achievable with the clinical formulation. (**A**) Mice were dosed orally with 20 mg/kg 4-HPR in three different formulations: a standard suspension formulation (HPMC-SV, 3 mL/kg), the MDE formulation (5% ethanol, 95% RPMI 1640 + 10% FBS; 8.8 mL/kg), and the HC formulation (92% corn oil, 8% polysorbate 80; 3 mL/kg). Data represent the mean ± SD of 3 mice per time point for the plasma concentration versus time. (**B**) Simulated exposure profiles in mice for b.i.d administration of 4-HPR at 20 mg/kg [19] for the formulations in (**A**). The higher plasma exposure of 4-HPR after dosing with either the MDE or HPMC-SV formulations, compared to HC, was statistically significant (see Appendix A).

Simulated exposure profiles for a twice-daily oral dosage of 4-HPR at 20 mg/kg in mice (as administered in Fraser et al. [19]) were modelled, revealing for the first time that the minimum plasma concentration for MDE under this dosage regime was approximately 1.0 μM (Figure 1B). At the same dose level, the HC formulation appeared to achieve therapeutic concentrations for a portion of the timescale in the model but showed a significantly lower trough concentration of approximately 0.3–0.4 μM.

### 3.2. 4-HPR Is Relatively Soluble in Mixed Medium-Chain Lipids In Vitro and Self-Emulsifying Lipid-Based Formulations Containing Them

Building on the above results for the HC formulation, we set out to develop novel formulations that may increase in vivo exposure of 4-HPR. Structurally derived from vitamin A, 4-HPR is strongly hydrophobic, with concomitant poor oral bioavailability due to low solubility and permeability [32], which in turn can result in problems with dose escalation in experimental tumor therapy due to the high number of capsules to be taken when delivered by conventional formulation approaches [26]. Lipid-based formulations have been investigated as a means to increase apparent gastrointestinal solubilisation, as well as benefit permeability and reduce the impact of first-pass metabolism of drugs with low aqueous solubility [33,34]. Lipid-based formulations can be classified into four main types based on the relative proportions of their constitutive lipid, surfactants and cosolvents excipient [30], with self-emulsifying drug delivery systems or SEDDS (Type II, Type IIIA and Type IIIB) spontaneously emulsifying with gentle mixing of the formulation with gastrointestinal fluids. To this end, the equilibrium solubility of 4-HPR in individual excipients was measured in a range of commonly used lipid-based formulation components (Table 2), including medium- and long-chain lipids, low- and high-HLB (hydrophobic lipophilic balance) surfactants and cosolvents. Additionally, 4-HPR was relatively highly soluble in mixed medium-chain lipids (e.g., Capmul MCM EP, 34 mg/g), cosolvent ethanol (113 mg/g), low-HLB surfactants (e.g., Lauroglycol 90, 39 mg/g) and high-HLB surfactants (Tween 85, 71 mg/g).

The results were used to guide the development of lipid-based formulations (Table 3) of primarily type III of the lipid formulation classification system [35]. Measured equilibrium solubility values were compared to predicted equilibrium solubility values (based on additive solubility in individual excipients), with the majority of formulations underperforming the predicted values. Some formulations were better able to solubilise 4-HPR than predicted (11, 13, 19, 23, 25, 26), with formulation candidates 23 (“N23”) and 25 (“N25”) selected for further study based on their high solubility and stability after 1:5 dispersion in water, and these formulations are analysed in more detail below in parallel with the corn oil-based HC and MDE formulations. The divergence between observed and predicted values here underlines the challenge of working with a complex molecule like 4-HPR, as well as the key importance of confirming predictions with direct experimental measurements in lipid-based formulation approaches.

### 3.3. Novel 4-HPR Formulations Retain Greater Solubilisation Than Existing Formulations after In Vitro Digestion and Increase In Vivo Exposure

To explore whether the developed self-emulsifying lipid-based formulations were able to retain drug solubilization during dispersion and subsequent digestion in the gastrointestinal tract, formulations were evaluated using an established in vitro model of lipid digestion [36], with samples assayed using the HPLC method (Section 2.5) [36,37,38]. The distribution of 4-HPR in the oil, aqueous and pellet phases of formulations N23, N25 and HC was analysed over time (see Figure 2). The HC (clinical) formulation was marked by virtually complete drug precipitation upon dispersion, whereas N23 and N25 both showed significant 4-HPR solubilisation after dispersion and markedly lower precipitation.

These results encouraged us to examine the systemic exposure of 4-HPR in male C57Bl/6 mice after oral administration of formulations N23 and N25 resulting in 4-HPR exposure that was comparable to the MDE formulation, both in terms of plasma C_max_ and AUC (Figure 3A, Table 4). Pre-dispersion of formulation N25 2:1 in water to achieve a matching dose volume of 3 mL/kg was shown not to have a significant effect on in vivo 4-HPR exposure compared to without pre-dispersion (Appendix A). Based on these results, the lipid-based formulation candidates yielded a substantial increase in 4-HPR exposure as compared to the HC formulation, offering an apparent absolute bioavailability of 17–18% and demonstrating comparable exposure profiles to the MDE formulation. In the case of formulations N23 and N25, the higher plasma exposure of 4-HPR compared to HC was statistically significant (see Appendix A); in the case of non-dispersed N25, the higher exposure of 4-HPR compared to the MDE formulation was also statistically significant (see Appendix A).

This suggests that both lipid-based formulations would be able to achieve comparable efficacy in preventing DENV infection in a mouse model of severe disease. Simulated exposure profiles for twice-daily oral dosage of 4-HPR at 20 mg/kg in mice revealed a trough concentration of approximately 1.2 μM for formulation N25 (Figure 3B), very similar to the results observed for the MDE formulation (see Figure 1B; Table 1).

### 3.4. In Vivo 4-HPR Exposure Is Limited by Both Solubility and Extensive First-Pass Intestinal Elimination

In light of the low absolute bioavailability of 4-HPR from both solubilised N23 and N25 formulations and the potential role of CYP450 enzymes in 4-HPR metabolism [39,40], we decided to explore the potential for first-pass metabolism to significantly limit 4-HPR oral bioavailability. To do this, 4-HPR was administered orally and intravenously in the presence of the pan-CYP inhibitor, 1-aminobenzotriazole (ABT) and compared to data in the absence of ABT to identify the impact of CYP inhibition.

Oral exposure of 4-HPR in formulation N25 in C57Bl/6 mice treated with 50 mg/kg ABT was almost 5-fold higher than that observed in mice that had not been pre-treated with ABT (Figure 4A, Table 5). In a similar manner, oral exposure of 4-HPR in corn oil-based HC formulation was increased 3-fold in mice treated with ABT. Formulation N25 resulted in approximately 4-fold greater exposure and almost 3-fold greater bioavailability (calculated relative to IV 4-HPR mice) than the HC formulation (Figure 4B, Table 5). These results suggest a major role for CYP450 in the first-pass metabolism of oral 4-HPR and demonstrate that a self-emulsifying lipid-based formulation approach has the ability to significantly increase 4-HPR exposure, with further increases in exposure possible through the use of a CYP450 inhibitor.

To define the contribution of first-pass hepatic metabolism to clearance, 4-HPR was administered intravenously, and the concentration-time profile and associated PK parameters were determined (Figure 5, Table 6) in the absence or presence of ABT. ABT treatment resulted in only a limited 0.2-fold increase in exposure, with very low in vivo systemic clearance (4.4 mL/min/kg) observed under both conditions. This low systemic clearance suggests that hepatic first-pass elimination would also be low, which further raises the possibility that oral bioavailability is limited by significant intestinal, possibly enterocyte-based metabolism.

Finally, to shed light on the basis for the differential effects of hepatic and intestinal CYP-mediated metabolism, plasma protein binding of 4-HPR was determined in both human and mouse plasma. These studies revealed that 4-HPR is > 99.95% bound to protein in both human and mouse plasma (Table 7). Thus, the differing contributions of hepatic and intestinal clearance of 4-HPR can potentially be attributed to the differences in unbound concentrations within the enterocyte and those found systemically due to the very high plasma protein binding of 4-HPR.

## 4. Discussion

Previous work identified 4-HPR as a potent anti-dengue drug and demonstrated efficacy in an in vivo model of lethal DENV infection when delivered as the MDE suspension formulation [19]. The present study analysed the PK properties of the MDE suspension formulation for the first time, with simulated exposure profiles for a twice-daily oral dosage of 4-HPR at 20 mg/kg in mice as administered in Fraser et al. [19], indicating that the minimum plasma concentration under this dosage regime is approximately 1.0 μM. Importantly, this suggests that a trough concentration of 1 μM is sufficient for antiviral activity, which parallels the in vitro EC50 of 1–2 μM for both DENV and ZIKV [9]. Carocci et al. [21] showed that b.i.d administration of 4-HPR (albeit at a much higher dose of 180 mg/kg) can reduce DENV2 viremia > 50-fold in an AG129 mouse viremia model.

Although the MDE formulation is not suitable for human use, and the standard corn oil-based human clinical (HC) suspension formulation shows >3 times lower exposure at the same dose (20 mg/kg) here in mice, it should be noted that human clinical trials have used doses as high as 4000 mg/m^2^/day with no significant toxicity (equivalent to approximately 100 mg/kg) [26]. This suggests that the HC formulation would be able to safely achieve a therapeutic dose exposure, but it also highlights the need for a formulation with improved performance to reduce issues associated with patient compliance due to large dose volumes.

Various approaches have been investigated to improve 4-HPR exposure in a cancer context [32,39,41,42,43,44,45,46]. This study describes two novel self-emulsifying lipid-based formulations of 4-HPR with relatively high solubility that appear to have superior exposure profiles to the HC formulation. Simulated exposure profiles for twice-daily oral dosage of 4-HPR at 20 mg/kg in mice reveal a trough concentration of approximately 1.2 μM for lipid formulation 25 (Figure 3B), thus allowing a 4-fold reduction in dosage to achieve the same exposure as the HC formulation. An alternative formulation using 4-HPR complexed with 2-hydroxypropyl-beta-cyclodextrin (nanofenretinide) similarly achieved higher (approximately 3-fold) exposure in mice than the HC formulation [41]. A phase I human clinical study in paediatric neuroblastoma patients described that 4HPR/LYM-X-SORB (a free-flowing powder matrix of lysophosphatidylcholine, monoglycerides and free fatty acids at a molar ratio of 1:4:2) [42], when mixed into a meal replacement drink, resulted in approximately 2-fold higher plasma concentration than that previously achieved at similar doses with the HC formulation [24,43]. Extrapolating to humans from our mouse PK data, formulations 23 and 25 may speculatively afford a potential improvement over the 4HPR/LYM-X-SORB formulation around 2-fold.

Intriguingly, the lipid-based as well as MDE formulations appear to encounter an upper limit on the bioavailability of less than 20%, with examinations of PK profiles of oral and IV 4-HPR in the absence or presence of the CYP-inhibitor, ABT, revealing a limited role for first pass hepatic elimination and implying significant intestinal metabolism. A likely mechanism for these differing contributions to first-pass elimination is the extremely high plasma protein binding of 4-HPR, which would attenuate the systemic hepatic clearance but may have a much lower effect on intestinal metabolism; this has not been previously reported. The data also lend support to the notion of pharmacologically modulating CYP-mediated metabolism of 4-HPR in patients through the use of drugs such as ketoconazole or ritonavir as a way of increasing 4-HPR exposure in existing formulations [24,43]. In this context, it will be important to consider drug interactions resulting from the use of CYP inhibitors, such as ritonavir, that have been well studied for various anti-human immunodeficiency virus-1 therapeutics [47,48], and more recently for nirmatrelvir (paxlovid) as an antiviral for SARS-CoV-2 [49]. It will also be critical to take into account previous isolated studies in cultured mosquito cells [50] and in an 81-year-old patient with a long history of hepatitis C infection [51] that imply that CYP inhibitors may themselves have the potential to enhance DENV replication.

In summary, this study establishes that the existing HC formulation for 4-HPR with b.i.d administration can achieve in vivo 4-HPR exposure levels necessary for effective dengue virus prophylaxis in a mouse model. It also describes a successful lipid-based formulation approach for 4-HPR with improved in vitro and in vivo PK properties in mice that may allow dose volumes to be reduced to help increase patient compliance. Importantly, co-administration with CYP inhibitors is eminently worthy of further consideration in terms of the potential enhancement of 4-HPR efficacy in both cancer and flavivirus clinical settings.

## 5. Patent

International PCT application PCT/AU2022/050091 pending: Jans, D.A., Wagstaff, K.M., Porter, C.J., Martin, A.J. Formulations for improved bioavailability of fenretinide. Monash University.

## Figures and Tables

**Figure 2 pharmaceutics-15-01974-f002:**
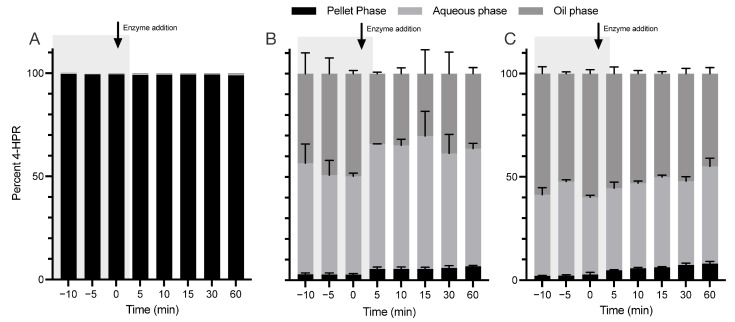
Novel isotropic SEDDS formulations N23 and N25 retain greater solubilisation of 4-HPR than the existing lipid suspension HC formulation after in vitro digestion. Drug distribution profiles after in vitro digestion for the HC ((**A**), 115 mg/g), N23 ((**B**), 39 mg/g) and N25 ((**C**), 55 mg/g) formulations. Results represent the mean ± SD (*n* = 3), with the exception of A, which represents a single typical experiment from a series of similar experiments.

**Figure 3 pharmaceutics-15-01974-f003:**
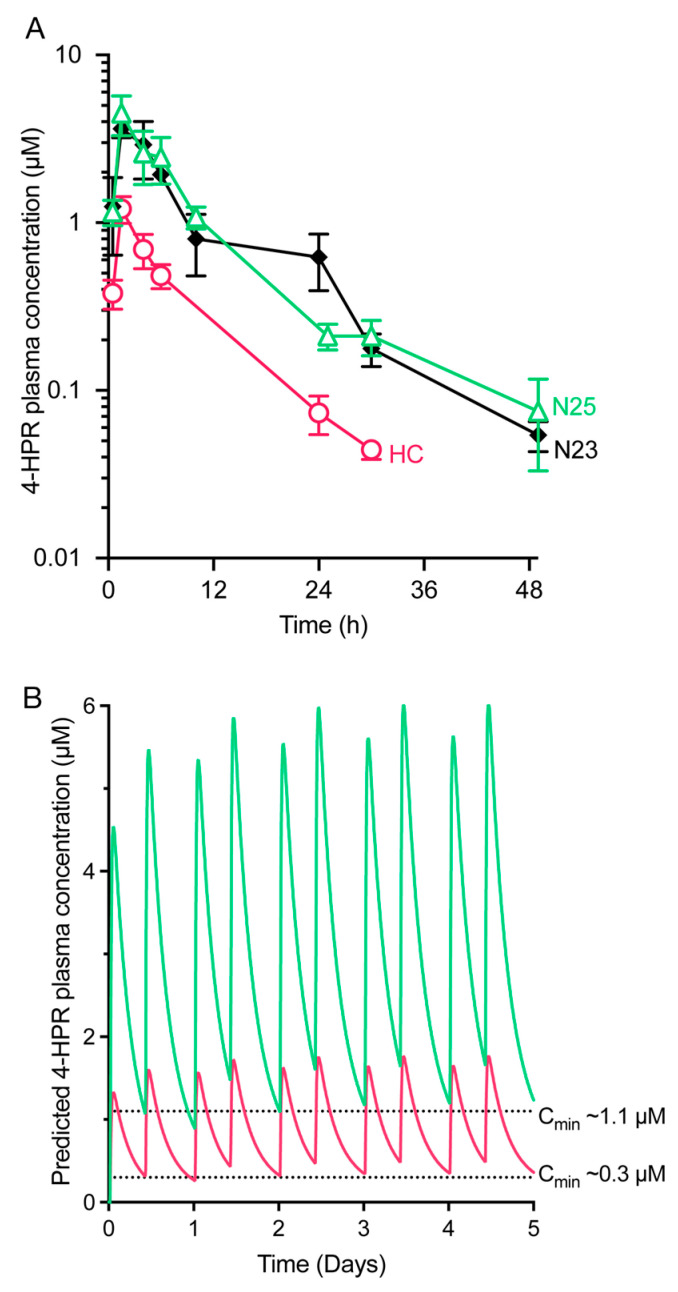
Novel lipid-based 4-HPR formulations can enhance in vivo exposure. (**A**) Mice were orally dosed with 20 mg/kg 4-HPR in self-emulsifying lipid-based formulations N23 (administered neat; 2 mL/kg) and N25 (pre-dispersed 1:2 (*v*/*v*) in water with vortex mixing; 3 mL/kg—see Appendix A) or the HC formulation (3 mL/kg). Data represent the mean ± SD of 3 mice per time point for the plasma concentration versus time. The higher plasma exposure of 4-HPR after dosing with either formulations N25 or N23 compared to HC was statistically significant (see Appendix A). (**B**) Simulated exposure profile in mice for b.i.d administration of 4-HPR at 20 mg/kg in formulation N25 compared to HC (see Figure 1B). Parameter estimates are shown in Appendix A.

**Figure 4 pharmaceutics-15-01974-f004:**
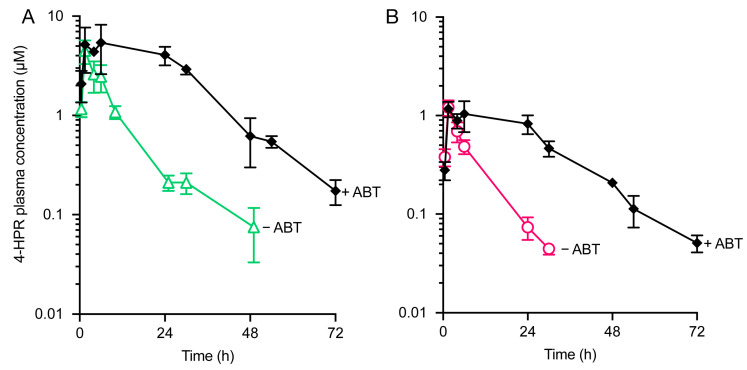
CYP inhibition by ABT enhances 4-HPR plasma exposure after oral dosage. Mice were dosed with 20 mg/kg 4-HPR in either formulation N25 (**A**) or HC (**B**), with and without concurrent ABT treatment at 50 mg/kg b.i.d. Data represent the mean ± SD of *n* = 3 mice per time point for the plasma concentration versus time.

**Figure 5 pharmaceutics-15-01974-f005:**
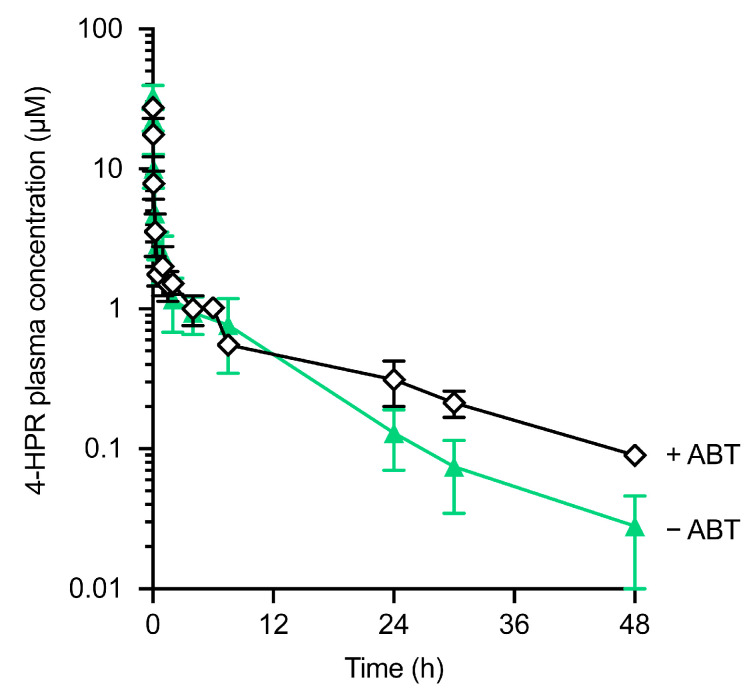
CYP inhibition by ABT only enhances 4-HPR plasma exposure to a small extent after IV dosage. Mice were dosed with 2 mg/kg 4-HPR IV via bolus tail vein injection (2 mL/kg), with and without concurrent ABT treatment at 50 mg/kg b.i.d. 4-HPR was administered and dissolved in a non-aqueous organic vehicle consisting of 10% DMSO and 90% PEG400. Data represent the mean ± SD of *n* = 6 mice per time point.

**Table 1 pharmaceutics-15-01974-t001:** Summary of PK analysis for 4-HPR in male C57Bl/6 mice following single-dose oral administration.

Parameter ^1^	Formulation		
	HC ^2^	HPMC-SV ^2^	MDE ^2^
Average Dose (mg/kg)	18.2	19.2	19.7
Dose Volume (mL/kg)	3	3	8.8
Apparent t_1/2_ (h)	6.9	7.2	7.3
Plasma AUC_0-inf_ (µM h)	10.3	12.7	35.1
Plasma C_max_ (µM)	1.21	1.00	4.80
Plasma T_max_ (h)	1.5	1.5	1.5
Bioavailability (%)	5.2	6.3	17

^1^ Mice were dosed as per Figure 1. ^2^ Results are based on the mean of *n* = 3 mice per time point.

**Table 2 pharmaceutics-15-01974-t002:** Equilibrium solubility of 4-HPR in individual excipients.

Excipient	Type of Excipient	4-HPR Concentration (mg/g) ^1^
Capmul-MCM-EP	Mixed medium-chain glycerides (lipid)	34 ± 4.3
Captex 355 EP/NF	Medium-chain triglyceride (lipid)	14 ± 0.1
Maisine 35-1	Mixed long-chain glycerides (lipid)	21 ± 7.6
Maisine CC	Mixed long-chain glycerides (lipid)	26 ± 1.6
Soybean Oil	Mixed long-chain glycerides (lipid)	2.7 ± 0.1
Corn oil	Mixed long-chain glycerides (lipid)	3.9 ± 0.2
PEG400	Cosolvent	82 ± 0.7
Transcutol	Cosolvent	217 ± 8.8
Ethanol	Cosolvent	113 ± 1.3
Capryol 90	Surfactant	48 ± 6.0
Lauroglycol 90	Surfactant	39 ± 1.4
Kolliphor RH 40	Surfactant	5.5 ± 0.6
Cremophor EL	Surfactant	10 ± 0.2
Labrasol	Surfactant	7.3 ± 0.1
Gelucire 44/14	Surfactant	6.0 ± 0.2
Tween 85	Surfactant	71 ± 9.1
Polysorbate 80	Surfactant	24 ± 4.9

^1^ Results are for the mean ± SD (*n* = 3).

**Table 3 pharmaceutics-15-01974-t003:** Equilibrium solubility of 4-HPR in self-emulsifying lipid-based formulations.

ID No.	LFC ^1^	Composition	4-HPR Solubility (mg/g)	
			Expected ^2^	Measured ^3^
1	IIIB LC	25% Maisine 35-1, 25% PEG400, 50% Kolliphor RH-40	29	5.1 ± 0.4
2	IIIA MC	25% Capmul MCM-EP, 25% Capryol 90, 50% Kolliphor RH-40	23	5.5 ± 1.7
3	IIIA LC	32.5% Maisine 35-1, 32.5% PEG400, 35% Kolliphor RH-40	35	4.4 ± 0.2
4		32.5% Capmul MCM-EP, 32.5% Capryol 90, 35% Kolliphor RH-40	29	6.0 ± 0.2
5	IV	50% Capryol 90, 50% Kolliphor RH-40	27	6.0 ± 0.2
6	IV	50% PEG400, 50% Kolliphor RH-40	44	10 ± 0.9
7	IIIA MC	32.5% Captex 355, 32.5% Capmul MCM-EP, 35% Kolliphor RH-40	18	3.4 ± 0.3
8	IIIA LC	32.5% Maisine 35-1, 32.5% Soybean Oil, 35% Kolliphor RH-40	10	2.0 ± 0.2
9	IIIB MC	25% Capmul MCM-EP, 50% Kolliphor RH-40, 25% Ethanol	61	10 ± 0.2
10	IIIB LC	25% Maisine 35-1, 25% Ethanol, 50% Kolliphor RH-40	58	10 ± 1.2
11		50% Capmul MCM-EP, 50% Capryol 90	41	68 ± 1.0
12		50% PEG400, 50% Capmul MCM-EP	58	5.4 ± 0.1
13		50% Capmul MCM-EP, 50% Tween 85	53	66 ± 4.4
15	IIIA LC	32.5% Maisine CC, 32.5% PEG400, 35% Tween 85	60	5.1 ± 0.1
18	IIIB LC	25% Maisine CC, 25% PEG400, 50% Tween 85	63	5.2 ± 0.1
19	IIIB LC	25% Maisine CC, 25% Ethanol, 50% Tween 85	70	85.3 ± 3.4
20		50% Capryol 90, 50% Tween 85	60	59.5 ± 1.9
N23	IIIA LC	50% Maisine CC, 10% Ethanol, 40% Tween 85	66	75.6 ± 1.0
N25	IIIA LC	25% Maisine CC, 25% Lauroglycol 90, 10% Ethanol, 40% Tween 85	56	83.8 ± 0.4
26	IIIA MC	25% Capmul MCM-EP, 25% Captex 355 EP/NF, 50% Tween 85	48	58.5 ± 3.3
27		55% Maisine CC, 45% Tween 85	46	34.1 ± 1.2
28		50% Masine CC, 50% Tween 85	48	32.4 ± 0.2
29		28.33% Maisine CC, 28.33% Lauroglycol 90, 43.33% Tween 85	49	38.9 ± 1.2
30		25% Maisine CC, 25% Lauroglycol 90, 50% Tween 85	51	37.1 ± 0.7
HC		92% Corn Oil, 8% Polysorbate 80	22	2.1 ± 0.1

^1^ Lipid formulation classification as per [36]. ^2^ Expected solubilities calculated additively based on individual excipient solubility results from Table 2. ^3^ Results for measured solubility values are for the mean ± SD (*n* = 3).

**Table 4 pharmaceutics-15-01974-t004:** Summary of PK analysis for 4-HPR in male C57Bl/6 mice following single-dose oral administration using novel lipid formulations.

Parameter ^1^	Lipid Formulation^2^	
	N23	N25 ^3^
Average Dose (mg/kg)	20.5	21.2
Dose Volume (mL/kg)	2	3
Apparent t_1/2_ (h)	9.5	10.5
Plasma AUC_0-inf_ (µM h)	36.6	38.7
Plasma C_max_ (µM)	3.63	4.50
Plasma T_max_ (h)	1.5	1.5
Bioavailability (%)	17	18

^1^ Mice were dosed as per Figure 3. ^2^ Results are based on the mean of *n* = 3 mice per time point. ^3^ Plot for non-dispersed data for formulation N25 is in Appendix A.

**Table 5 pharmaceutics-15-01974-t005:** Summary of PK analysis for 4-HPR in male C57Bl/6 mice following single-dose 20 mg/kg oral administration of formulations N25 or HC in the presence or absence of CYP inhibitor ABT (50 mg/kg, b.i.d).

**Oral Administration of 4-HPR in N25**		
**Parameter ^1^**	**−ABT ^2^**	**+ABT ^2^**
Average Dose (mg/kg)	21.2	22.7
Apparent t_1/2_ (h)	10.5	12.6
Plasma C_max_ (µM)	4.5	5.4
T_max_ (h)	1.5	6
Plasma AUC_0-inf_ (µM h)	38.7	174
Plasma CL/F (mL/min/kg)	23.3	5.5
Apparent BA (%)	18	66
**Oral Administration of 4-HPR in HC**		
**Parameter** ** ^1^ **	**−ABT** ** ^2^ **	**+ABT** ** ^2^ **
Average Dose (mg/kg)	18.2	18.7
Apparent t_1/2_ (h)	6.9	12
Plasma C_max_ (µM)	1.21	1.17
T_max_ (h)	1.5	1.5
Plasma AUC_0-inf_ (µM h)	10.3	35.4
Plasma CL/F (mL/min/kg)	75.6	22.5
Apparent BA (%)	5.4	16.0

^1^ Mice were dosed as per Figure 4. ^2^ Results are based on the mean of *n* = 3 mice per time point; data for formulation N25 −ABT are repeated from Table 4.

**Table 6 pharmaceutics-15-01974-t006:** Summary of PK analysis for 4-HPR in male C57Bl/6 mice following single-dose 2 mg/kg IV administration in the presence or absence of CYP inhibitor ABT (50 mg/kg, b.i.d).

IV Administration of 4-HPR		
Parameter ^1^	−ABT ^2^	+ABT ^2^
Average dose (mg/kg)	2.1	2.1
Apparent t_1/2_ (h)	8.1	12.1
Plasma AUC_0-inf_ (µM h)	21.6	24.5
Plasma CL (mL/min/kg)	4.1	3.6
Plasma V_SS_ (L/kg)	1.9	3.2

^1^ Mice were dosed as per Figure 5. ^2^ Results are based on the mean of *n* = 6 mice per time point.

**Table 7 pharmaceutics-15-01974-t007:** Estimation of the extent of 4-HPR protein binding in human and mouse plasma.

Plasma Species	Fraction Unbound 4-HPR (% Bound) ^1^
Human	<0.0005 (>99.95% bound)
Mouse	<0.0005 (>99.95% bound)

^1^ Results for are presented as the mean ± SD of *n* = 3–4 independent dialysis units.

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
