# Peer review of "Increased In Vivo Exposure of N-(4-Hydroxyphenyl) Retinamide (4-HPR) to Achieve Plasma Concentrations Effective against Dengue Virus"

_pharmaceutics, 2023, doi:10.3390/pharmaceutics15071974_

Round 1

Reviewer 1 Report

The study entitled “Increased in vivo exposure of N-(4-hydroxyphenyl) retinamide 2 (4-HPR) to achieve plasma concentrations effective against dengue virus” by Martin et al., deals with the antiviral activity of fenretinide. This drug has been studied for a long time mainly for its anticancer activity.  Indeed this drug induces few side effects and no induction of resistance, however its poor solubility strongly limits its efficacy in vivo. The Authors here provide a PK study in mice. The Introduction needs some adjustments as it lacks completely of the description of formulations already present in the literature and also the approach used by the Authors (SDDS)  is barely mentioned and introduced. Overall the description of various formulations employed are not always clear and not reported in detail in the M&M.

Specific comments:

-        Line 79, the Authors should report some of the attempts already made to enhance fenretinide solubility and bioavailability, as several drug delivery systems have been already developed (10.3390/ph16030388;  10.1016/j.jconrel.2013.06.015)

-        Line 84, the Author stated “In this study, we set out to evaluate the in vivo exposure profile of the 4-HPR formulation (mouse dengue efficacy formulation - MDE) shown to be effective in severe DENV infection [19]”, it not clear to what formulation they refer as in ref 19 a formulation is not described

-        Line 87, check the meaning

-        Line 82, the last paragraph of the introduction is not well structured and organized. The Authors should firstly conclude the description of their formulative approach, pointing out the reasons of their selection. In addition, the results of the research reported should be deleted.

-        Line 118, the reader cannot understand what formulation is administered to mice. What is the formulation given per os and IV? Please provide a detailed description.

-        The Authors should add a subsection dedicated to statistical analysis and evaluate if there are significant differences among the novel formulation and the control formulation (Figure 4 and 3).

The English language is basically correct

Author Response

We thank the Reviewer for the positive comments and important suggestions. The changes we have made to the manuscript in response to the Reviewer’s comments are as follows.

  1. SEDDS is now explained with a paragraph and references (including new References 32-34) at the start of Section 3.2 (Lines 290-301 - see also response to Specific Comment Line 82). We thank the Reviewer for encouraging us to include this.
  2. The formulations were all documented in the Tables in the original manuscript, but we now include Supplementary Table S1 (referred to in Section 2.2, lines 131-2) which has the specifics of all of the formulations used in the PK experiments. We thank the Reviewer for this valuable suggestion (see also response to Specific Comment e).
  3. Specific comments:
    1. Line 79 - we thank the Reviewer ! – the original manuscript included a number of examples of previous work to enhance fenretinide exposure in the context of cancer treatment in the Discussion, but we have now expanded this to include a number of additional studies (including one from 2023 which post-dated our manuscript drafts – we are very happy to have been alerted by the Reviewer to include this !) in Lines 443-458 (with inclusion of new References 32, 45 and 46). We are indebted to the Reviewer.
    2. Line 84 – this refers to the MDE formulation – this is now clearly stated in the manuscript.
    3. Line 87 – we have amended the text as requested.
    4. Line 82 – we respectfully point out that our manuscript has 3-fold aims (one is to estimate the efficacious therapeutic serum concentration of 4-HPR for protection used in Publication 19, the second is to develop new formulations to increase 4-HPR exposure [the approach taken is explained in Section 3.2], and the third is to perform experiments to investigate the pathways of 4-HPR metabolism). Further, “flagging” in quintessential form the multiple different research aspects of the study to put the detailed part of the paper (Methods/Results/Discussion) into context in the final paragraph of the Introduction is common practice in MDPI journals. We thank the Reviewer.
    5. Line 118 – the formulation is now clearly defined in Supplementary Table S1. We thank the Reviewer.
    6. We thank the Reviewer (and also Reviewer 3) for the suggestion to include statistical analysis. The analysis approaches are now documented in Material and Methods Section 2.4 second paragraph, and presented in Supplementary Table S3. Excitingly, there is a significant difference in the exposure afforded by the new formulations compared to the control HC formulation – we thank the Reviewer again for encouraging us to do this analysis confirming the significant improvement afforded by formulation N25 in particular.

We thank the Reviewer again for his invaluable input into our manuscript !

David Jans

Reviewer 2 Report

This is an interesting manuscript about the development of the known drug 4-HPR as a possible orally administered treatment option for the therapy of Dengue virus infections. The manuscript is well written and designed. It shows possible formulation options to reach the goal of an orally active 4-HPR drug in the future. I recommend acceptance after minor revision:

Section 3.2., line 312: Please mention the 18.5% bioavailability for N25 here in the main text.

Discussion: The authors correctly stated that co-drugs such as ketoconazole and ritonavir can reduce the CYP-based metabolism of drugs. Please mention Paxlovid as a prominent antiviral drug example here (and/or further examples).

Discussion: There is literature of ritonavir as a possible co-drug for Dengue treatment (e.g., PMID 27893666). In addition, it was shown that ketoconazole can enhance Dengue viral replication (PMID 10725192) and, thus, might be less suitable for a combination with Dengue antivirals. Please cite and discuss.

Author Response

We thank the Reviewer for the positive comments and important suggestions. The changes we have made to the manuscript in response to the Reviewer’s comments are as follows.

Minor revision:

  1. Section 3.2 – we have included the bioavailability for N23 and N25 as requested (revised manuscript lines 338-340).
  2. Discussion – as requested, we now mention Paxlovid and HIV-1 antivirals in this context in the Discussion as appropriate (Lines 471-473 with new references 47-49). We thank the Reviewer for this suggestion.
  3. Discussion – the two references flagged by the Reviewer are now cited (new references 50,51), with the important caveat now pointed out in Lines 473-476. We thank the Reviewer for this important suggestion.

We thank the Reviewer again for his/her invaluable suggestions to improve the manuscript.

David Jans

Reviewer 3 Report

The article by Alexander Martin and co-authors corresponds to the theme of the journal Pharmaceutics. It presents data on a studies of the promising antiviral compound N-(4-hydroxyphenyl) retinamide (4-HPR, or fenretinide). The results of research on the development of a convenient formulation of this compound and its pharmacokinetics are presented. These data may be of interest to researchers involved in the problems of antiviral drugs and medicinal chemistry.

The obtained results suggest that although the HC formulation achieves  plasma 4-HPR levels above the predicted minimum effective plasma concentration for anti-DENV activity in a mouse model, self-emulsifying lipid-based formulations and CYP inhibition represent viable future options to increase exposure. These results may be useful in the development of clinical applications of the title compound.

The article is well structured and clearly written. The literature references and the introduction take into account the previous results of the 4-HPR study.

According to the reviewer, the article can be recommended for publication after taking into account the following minor corrections:

1. It is recommended to include in the text of the article an illustration showing the chemical structure of the compound under study.

2. In the experimental part, it is recommended to add information about whether the results were statistically processed and, if so, how.

3. Table 3: It is recommended to explain the possible reasons for the strong deviation of some measured equilibrium solubility values from the predicted equilibrium solubility values.

English language is fine.Only minor editing of English language may be required

Author Response

We thank the Reviewer for the positive comments and important suggestions. The changes we have made to the manuscript in response to the Reviewer’s comments are as follows.

Minor Corrections:

  1. The chemical structure of 4-HPR is now included in Figure 1 – we thank the Reviewer for the suggestion.
  2. Statistical analysis has been added to the manuscript (Section 2.4; Supplementary Table S3) – we thank the Reviewer (and of course Reviewer 1).
  3. We thank the Reviewer –4-HPR is clearly a complex hydrophobic molecule ! We have added text to the end of Section 3.2 (Lines 303-306).

We thank the Reviewer again for the invaluable contribution.

David Jans

Round 2

Reviewer 1 Report

No more comments are needed.